# The m6A(m)-independent role of FTO in regulating WNT signaling pathways

Hyunjoon Kim[1,2],*, Soohyun Jang[1,2], Young-suk Lee[3],*

FTO and ALKBH5 are the two enzymes responsible for mRNA demethylation. Hence, the functional study of FTO has been focused on its mechanistic role in dynamic mRNA modification, and how this post-transcriptional regulation modulates signaling pathways. Here, we report that the functional landscape of FTO is largely associated with WNT signaling pathways but in a manner that is independent of its enzymatic activity. Re-analyses of public datasets identified the bifurcation of canonical and noncanonical WNT pathways as the major role of FTO. In FTO-depleted cells, we find that the canonical WNT/$\beta$-Catenin signaling is attenuated in a non-cell autonomous manner via the up-regulation of DKK1. Simultaneously, this up-regulation of DKK1 promotes cell migration via activating the noncanonical WNT/PCP pathway. Unexpectedly, this regulation of DKK1 is independent of its RNA methylation status but operates at the transcriptional level, revealing a noncanonical function of FTO in gene regulation. In conclusion, this study places the functional context of FTO at the branch point of multiple WNT signaling pathways and extends its mechanistic role in gene regulation.

## Introduction

Fat mass and obesity-associated (*FTO*) gene is associated with body mass index, obesity risk and type 2 diabetes (Frayling et al, 2007; Sanghera et al, 2008). Loss-of-function mutations of FTO also cause severe growth retardation and reduced brain volume (Boissel et al, 2009; Caglayan et al, 2016; Daoud et al, 2016). *FTO* is also one of the six genes deleted in Fused toe (Ft) mouse mutant (van der Hoeven et al, 1994; Peters et al, 1999). Despite growing evidence that FTO is implicated in a broad range of human disease, the exact physiological role of FTO is largely unknown.

FTO is an $\alpha$-ketoglutarate–dependent dioxygenase that demethylates mRNAs, tRNAs, and snRNAs to regulate RNA stability and function (Jia et al, 2011; Alarcon et al, 2015; Liu et al, 2015; Mauer et al, 2017; Wei et al, 2018). In particular, FTO demethylates and thereby stabilizes the mRNA of *MYC*, *CEBPA*, *ASB2*, and *RARA* leading to the enhanced oncogenic activity in certain leukemia cells (Su et al, 2018). FTO also plays a role in gluconeogenesis and thermogenesis by targeting the *FOXO1* mRNA (Peng et al, 2019). During motile ciliogenesis, we discovered that FTO demethylates the exon 2 of *FOXJ1* mRNA to enhance RNA stability (Kim et al, 2021). Much effort has been made to understand the molecular mechanisms of FTO, yet the cellular functions and upstream regulators of FTO that ultimately manifest its diverse roles warrants further investigation.

WNT signaling pathways regulate a variety of biological processes in development and tissue homeostasis (Logan & Nusse, 2004). In the canonical WNT/$\beta$-Catenin signaling pathway, the Wnt ligand (e.g., WNT3a) binds to the transmembrane receptor Frizzled (FZD) and low-density lipoprotein receptor-related 5/6 (LRP5/6) to stabilize $\beta$-Catenin proteins mainly by glycogen synthase kinase-3 (GSK3) inhibition (Piao et al, 2008; Taelman et al, 2010). Then, the $\beta$-Catenin proteins with TCF/LEF transcription factors activate gene expression of specific targets to regulate cell proliferation and differentiation (MacDonald et al, 2009). In the non-canonical WNT/PCP signaling pathway, Wnt ligands (e.g., WNT5A and WNT11) binds to the FZD receptor and co-receptors (e.g., ROR or Ryk) to activate small GTPases such as Rho, Rac, and Cdc42 and the c-Jun N-terminal kinase (JNK) (Lai et al, 2009). In addition to the cytoskeleton reorganization by the small GTPases, JNK phosphorylates c-Jun and the activated c-Jun along with transcription factors ATF2 and ATF4 orchestrate an intricate transcriptional network to regulate cell motility and tissue polarity (Lai et al, 2009). The other $\beta$-Catenin–independent, non-canonical WNT/Ca$^{2+}$ pathway induces the G-protein coupled receptor signaling cascade to increase intracellular calcium levels and regulate calcium/calmodulin-dependent protein kinases (Niehrs, 2012; Katoh, 2017). It is worth mentioning that canonical WNT signaling also leads to the cytoplasmic stabilization of many other GSK3-target proteins besides $\beta$-Catenin, termed the WNT/STOP pathway, thus remodeling the proteomic landscape (Acebron et al, 2014; Kim et al, 2015; Ploper et al, 2015).

In this study, we discovered an m6A-independent link between FTO and multiple WNT signaling pathways. Functional genomics analysis of FTO-depleted cells revealed specific down-regulation of canonical WNT and up-regulation of noncanonical WNT/PCP

[1]Center for RNA Research, Institute for Basic Science, Seoul, Korea   [2]School of Biological Sciences, Seoul National University, Seoul, Korea   [3]Department of Bio and Brain Engineering, Korea Advanced Institute of Science and Technology (KAIST), Daejeon, Korea

Correspondence: xenopus.h.kim@gmail.com; youngl@kaist.ac.kr
*Corresponding authors

signalings. We further find that the canonical WNT/β-Catenin pathway is attenuated in FTO-depleted cells, but in a m6A-independent manner. Among known WNT inhibitors, only DKK1 is up-regulated after FTO-depletion and attenuates canonical WNT activity in a non-cell autonomous manner. This FTO-dependent DKK1 up-regulation is also responsible for the enhanced cell migration of FTO-depleted cells presumably by activating the noncanonical WNT/PCP pathway. Overall, our results discover an m6A-independent role of FTO in regulating multiple WNT signaling pathways which may underlie the multifaceted complexity of FTO in human disease and development.

## Results

### Gene set enrichment analysis hints at functional interactions between FTO and WNT signaling pathways

To investigate the functional role of FTO, we strategically re-analyzed a published FTO and ALKBH5 siRNA knockdown and RNA-Seq dataset (Mauer et al, 2017). Both FTO and ALKBH5 are the responsible enzymes for RNA N6-methyladenosine (m6A) deme-thylation (Jia et al, 2011; Zheng et al, 2013). m6A is the most abundant internal mRNA modification (Desrosiers et al, 1974), and thus FTO and ALKBH5 may generally regulate a common set of biological processes. To investigate the FTO-specific biological functions, we therefore used the ALKBH5 siRNA knockdown RNA-Seq dataset as a negative control for our gene set enrichment analysis. Specifically, we compared the gene set enrichment scores of differentially expressed genes in FTO-depleted cells against those in ALKBH5-depleted cells (Figs 1A and S1A). Of note, negative enrichment scores indicate that genes of the set (e.g., Wnt signaling) are overall down-regulated after siRNA treatment.

WikiPathway gene set analysis (275 gene sets) revealed negative enrichment of "Cytoplasmic Ribosomal Proteins (WP477)" and "Amino Acid metabolism (WP3925)" for both FTO- and ALKBH5-depleted cells (Fig 1A). Interestingly, "Wnt Signaling (WP428)," "Mesodermal Commitment (WP2857)," and "Endoderm Differenti-ation (WP2853)" were only down-regulated in FTO-depleted cells (Fig 1A). Moreover, gene ontology enrichment analysis (3,091 gene sets) resulted in a negative enrichment of early developmental and cell proliferation-related biological processes specifically in FTO-depleted cells (Fig S1A). WNT signaling is one of the major signaling pathways governing cell proliferation and early developmental processes such as endomesodermal specification (Zorn et al, 1999), indicating that FTO might be coupled with WNT signaling pathways.

To test whether this perturbation of WNT signaling at the pathway level directly impacts the transcriptional landscape, we examined the change in mRNA level of known transcriptional targets of canonical and noncanonical WNT signaling pathways (Fig 1B, P-value < 0.05). Targets of the canonical WNT/β-Catenin tran-scription factors (i.e., TCF7 and LEF1) including MYC, VEGFA, and LEF1 were down-regulated in FTO-depleted 293T cells. This result indi-cates that the observed WNT disruption in FTO-depleted cells may lead to the inhibition of its targets, particularly that of the canonical WNT/β-Catenin signaling pathway. Of note, this is consistent with a

previous study which reported the inhibition of canonical WNT/β-Catenin signaling in Fto knockout mouse embryonic fibroblasts and in FTO-depleted 293T cells (Osborn et al, 2014).

Unlike the canonical WNT targets, targets of WNT/PCP pathway transcription factors (i.e., ATF2 and ATF4) such as CDKN1A, FOS, and HMOX1 were up-regulated, suggesting that FTO may inhibit the noncanonical WNT/PCP pathway. In fact, FTO-depleted 293T cells exhibit a statistically significant bifurcation of gene expression between the TCF7/LEF1 targets and ATF2/ATF4 targets (Fig 1C, P-value = 0.0081). A similar trend was also observed in other human cells (i.e., HepG2 and K562) after FTO depletion (Fig S1B), hinting that this specific coupling between FTO and distinct WNT signaling pathways might be common in multiple cellular contexts. It is worth emphasizing that this analysis does not demonstrate the context-independent connection of FTO with a particular WNT signaling pathway (e.g., the canonical WNT/β-Catenin pathway). Instead, this analysis hints that the specific functional role of FTO in contracts to ALKBH5 is associated with one or maybe multiple WNT signaling pathways depending on the physiological context.

### Loss of FTO inhibits the canonical WNT/β-catenin signaling pathway in a non-cell autonomous manner by up-regulating DKK1 mRNA expression

To investigate the molecular mechanism of how FTO regulates the WNT/β-Catenin signaling pathway, we used the TOP-Flash reporter harboring multiple TCF-binding sites and measured the tran-scriptional WNT/β-Catenin signaling activity. FTO knockdown in HeLa cells attenuated WNT3A-induced activation of TOP-Flash reporter but not the FOP-Flash reporter (Fig 2A) which is consis-tent with our bioinformatics result. This trend was found in multiple shRNAs targeting different sequences in FTO (Fig 2A), excluding the possibility of off-target effects. Moreover, knockout of FTO, but not ALKBH5, in HeLa cells using CRISPR/Cas9 system also attenuated the WNT3A-induced TOP-Flash reporter activation (Fig S2A), sup-porting the specific role of FTO in contrast to ALKBH5 in WNT signaling. The lack of accumulation of β-Catenin (Figs 2B and S2B) and attenuation of protein expression of WNT/β-Catenin targets such as AXIN2 and C-MYC (Fig S2C) after FTO depletion altogether indicates that FTO is required for the canonical WNT/β-Catenin signaling pathway.

Interestingly, similar results were also found in HEK293T cells (Fig S2D and E) but not in HepG2 cells (Fig S2F and G). Unlike HeLa and HEK293T cells, HepG2 cells express a constitutively active form of β-Catenin and used as a model system to study the active status of the canonical WNT/β-Catenin signaling pathway (de La Coste et al, 1998). Thus, this seemingly conflicting result in HepG2 cells may instead hint at the possibility that FTO modulates the canonical WNT signaling upstream of β-Catenin. To test this hypothesis, we co-cultured FTO-depleted HeLa cells and WT HeLa cells expressing the TOP-Flash reporter. We found decreased WNT3A signaling activity even when the WNT reporter-expressing control HeLa cells were co-cultured with FTO-depleted cells (Fig 2C), demonstrating non-cell autonomous inhibition of canonical WNT signaling by FTO-depleted cells. Culture medium (CM) from FTO-depleted cell also attenuated WNT3A-induced TOP-Flash reporter activation (Fig 2D) and β-Catenin accumulation (Fig 2E), indicating that secreted

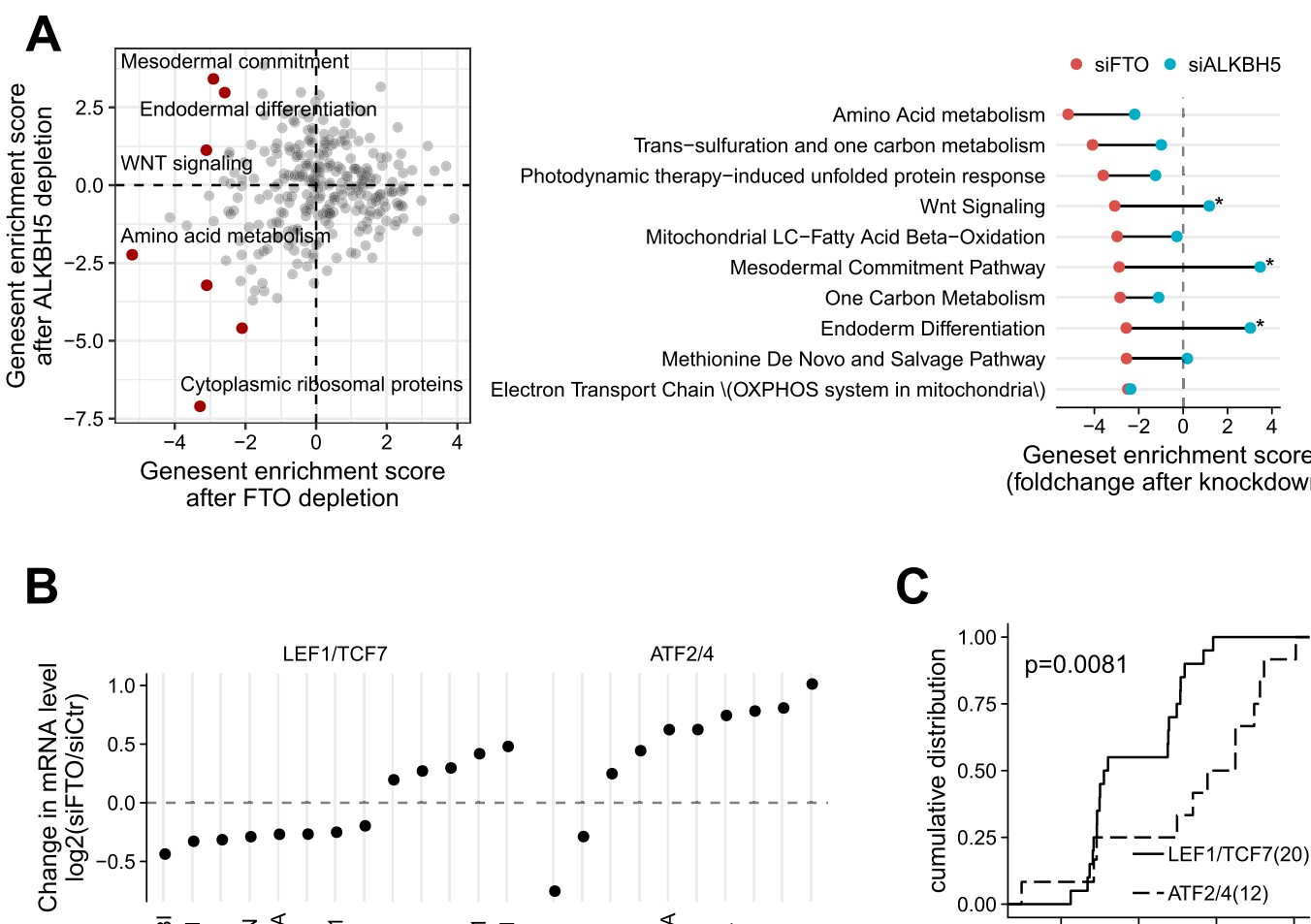

**Figure 1. Systems approach unravels potential interaction of FTO with WNT signaling pathways.**
**(A)** Enrichment score of 275 WikiPathway gene set pathways in FTO-depleted and ALKBH5-depleted 293T cells. Notable pathways are marked in red (left). Top 10 WikiPathway gene sets with greater negative enrichment score in siFTO-depleted cells (red dots) than in siALKBH5-depleted cells (blue dots). Gene sets with enrichment score difference greater than four are marked (right). **(B)** Regularized fold change after FTO knockdown of differentially expressed genes (*P*-value < 0.05) of LEF1/TCF7 targets and ATF2/ATF4 targets. **(C)** Cumulative plot of LEF1/TCF7 targets (solid) and ATF2/ATF4 targets (dashed) after FTO knockdown. The number of genes is shown in parenthesis. One-sided Kolmogorov–Smirnov test was used to test statistical significance.

factors from FTO-depleted cells antagonize the canonical WNT signaling activity. Notably, only FTO-depleted cells displayed non-cell autonomous WNT inhibition, whereas METTL3- or YTHDF2-depleted cells showed no such activity (Fig 2C–E).

Unexpectedly, we found that WNT3A stimulation leads to the accumulation of FTO proteins in multiple cell lines (Figs 2B and S2B, C, E, and G), suggesting that the canonical WNT pathway may also regulate the function of FTO. *FTO* mRNA levels were not affected after WNT3A stimulation (Fig S2H), excluding the possibility of FTO as a transcriptional target of the canonical WNT pathway. Canonical WNT signaling is known to stabilize proteins other than $\beta$-Catenin by inhibiting GSK3 kinase (Taelman et al, 2010; Acebron et al, 2014; Kim et al, 2015), collectively known as the WNT/(STOP) STabilization Of Proteins pathway. Whereas FTO was not reported as a target of the WNT/STOP pathway, it was recently reported that GSK3 can

phosphorylate FTO and generate phosphodegron for polyubiquitination and subsequent proteasomal degradation (Faulds et al, 2018). Consistent with previous reports, both WNT3A stimulation and GSK3 inhibition (i.e., SB216763) induce FTO protein stabilization (Fig S2I) by inhibiting its polyubiquitination (Fig S2J), indicating that the canonical WNT/STOP signaling also stabilizes the FTO protein via GSK3 inhibition.

To identify the FTO-dependent factor(s) that antagonizes the WNT/$\beta$-Catenin signaling pathway, we measured the mRNA levels of known extracellular WNT antagonists. Only *DKK1* mRNA was up-regulated in FTO-depleted HeLa cells (Figs 3A and S3A and B). Secreted DKK1 protein level was also substantially increased in FTO-depleted HeLa cells (Figs 3B and S3C–E), suggesting that DKK1 may be the responsible factor for the non-cell autonomous WNT regulation in FTO-depleted cells. Notably, ALKBH5-depleted HeLa

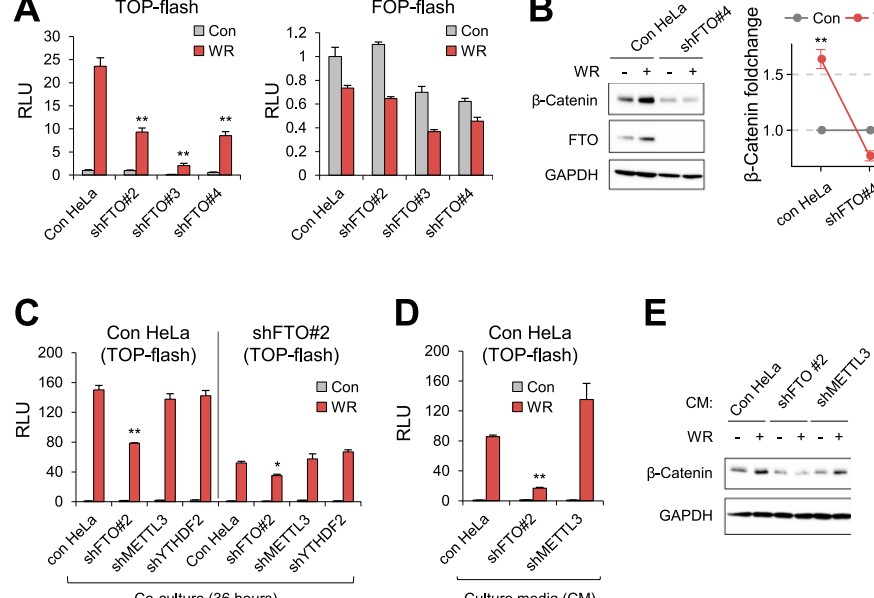

**Figure 2. Loss of FTO suppresses canonical Wnt/β-Catenin signaling.**
**(A)** HeLa cells stably expressing shRNAs targeting different sequences in FTO (#2, #3, #4; see the Materials and Methods section) were transfected with WNT signaling reporter (TOP-flash) or control reporter (FOP-flash) (n = 3). Relative Luciferase activities (RLU) were measured after 8 h of WNT stimulation. **(B)** Control HeLa cells or shFTO#4-expressing HeLa cells were stimulated with WR for 8 h and protein levels of β-Catenin and FTO were assessed by Western blot (n = 3). Note the stabilization of FTO protein by WNT signaling. **(C)** Control HeLa cells or shFTO#2-HeLa cells were transiently transfected with WNT reporter (TOP-flash) and co-cultured with cells expressing different shRNAs for 36 h as indicated (n = 3). Relative luciferase activities (RLU) were measured after 16 h of WNT stimulation. **(D)** Culture medium of HeLa cells transiently expressing WNT reporter (TOP-flash) was replaced with the medium harvested from control HeLa, shFTO#2-HeLa, or shMETTL3-HeLa before WNT stimulation for 16 h. RLU, Relative luciferase units (n = 3). **(E)** Cell lysates from Fig 2D were analyzed by Western blot for β-Catenin expression. GAPDH was used as a loading control for Western blot. WR, WNT3A, and R-Spondin1. Error bars are ± SEM. One-sided t test, **P < 0.01, *P < 0.05.

cells did not show increased *DKK1* mRNA levels (Fig S3B) nor secreted protein levels (Fig S3D and E). When DKK1 was transiently depleted with siRNA treatment, the culture medium from FTO-depleted HeLa cells no longer attenuated the WNT3A-induced TOP-Flash reporter activation (Fig 3C), indicating that DKK1 plays a role in FTO-dependent regulation of the canonical WNT signaling pathway. Similar results were found after neutralization of the DKK1 protein in the extracellular milieu by anti-DKK1 antibody (Fig S3F). The culture medium (CM) of WNT-active HepG2 cells also attenuated WNT activation in HeLa cells and the immunodepletion of DKK1 from the CM abrogated this inhibitory activity (Fig S3G), suggesting that this FTO-dependent DKK1 regulation is common in multiple cell systems. Altogether, these results demonstrate that secreted DKK1 proteins are responsible for the non-cell autonomous inhibition of canonical WNT/β-Catenin signaling in FTO-depleted cells. Notably, *DKK1* mRNA was also up-regulated in WNT-active HCT-8 cells (Fig 3D–F), excluding the possibility of β-Catenin–dependent up-regulation of DKK1 in FTO-depleted cells.

FTO demethylates N6-methyladenosine (m6A) and N6, 2′-O-dimethyladenosine (m6Am) of mRNAs and noncoding RNAs (Wei et al, 2018). m6Am has been proposed to stabilize mRNA and/or regulate cap-dependent translation (Mauer et al, 2017; Akichika et al, 2019). However, no substantial changes in m6A/m6Am modifications of *DKK1* mRNAs were observed in FTO-depleted HeLa and HCT-8 cells (Fig 3D). Moreover, the stability of *DKK1* mRNA did not increase after FTO knockdown (Fig S3H and I), altogether excluding the possibility of FTO-dependent stabilization of DKK1 mRNA via RNA demethylation. DKK1 mRNA levels were rescued in both wild-type and catalytic mutant FTO experiments (Fig 3G) along with its response to WNT/β-Catenin signaling (Fig S3J). Therefore, loss of FTO up-regulates DKK1 expression levels independent of its RNA demethylation activity.

On the contrary, *DKK1* pre-mRNA levels increased in FTO-depleted HeLa cells (Figs 3F and S3B), indicating that the loss of

FTO leads to the up-regulation of *DKK1* transcription. Consistently, Pol II occupancy near the *DKK1* transcription start site increased after FTO depletion (Fig S3K). *DKK1* mRNA and pre-mRNA levels were also elevated in FTO-depleted HCT-8 and HepG2 cells harboring constitutively activated WNT/β-Catenin signaling (Figs 3D and E and S3L). Culture medium from FTO-depleted HepG2 cells exhibited WNT inhibitory activity in a DKK1-dependent manner (Fig S3G). Of note, mRNA of canonical WNT target *AXIN2* was not up-regulated after FTO depletion (Fig 3H). These results exclude the possibility that the increased DKK1 expression in FTO-depleted cells is via WNT/β-Catenin-mediated transcriptional activation (Niida et al, 2004).

## Loss of FTO enhances cell migration via WNT/PCP pathway activation

In addition to the down-regulation of canonical WNT/β-Catenin signaling in FTO-depleted cells, we also observed a consistent up-regulation of the noncanonical WNT/PCP target genes across multiple cellular contexts (Fig 1B and C). WNT/PCP signaling is responsible for cell motility and polarity via Rho, Rac, and JNK activation (Steffen et al, 2013).

To investigate whether and how FTO regulates WNT/PCP pathway, we first performed an in vitro scratch wound healing assay to test whether there are any changes in cell motility and polarity in FTO-depleted cells. We found enhanced wound healing activity in FTO-depleted cells but not in ALKBH5-depleted cells (Fig S4A–C), indicating that FTO may play a specific role in cell migration. Moreover, FTO knockdown induced cell shape polarization at the wound edge (Fig 4A) but also in normal growth state (Figs 4A and S4D), suggesting that the FTO-dependent change in cell behavior is intrinsic rather than triggered by damage response signaling during wound healing process. Of note, FTO knockdown does not seem to promote epithelial–mesenchymal transition per se as evident by

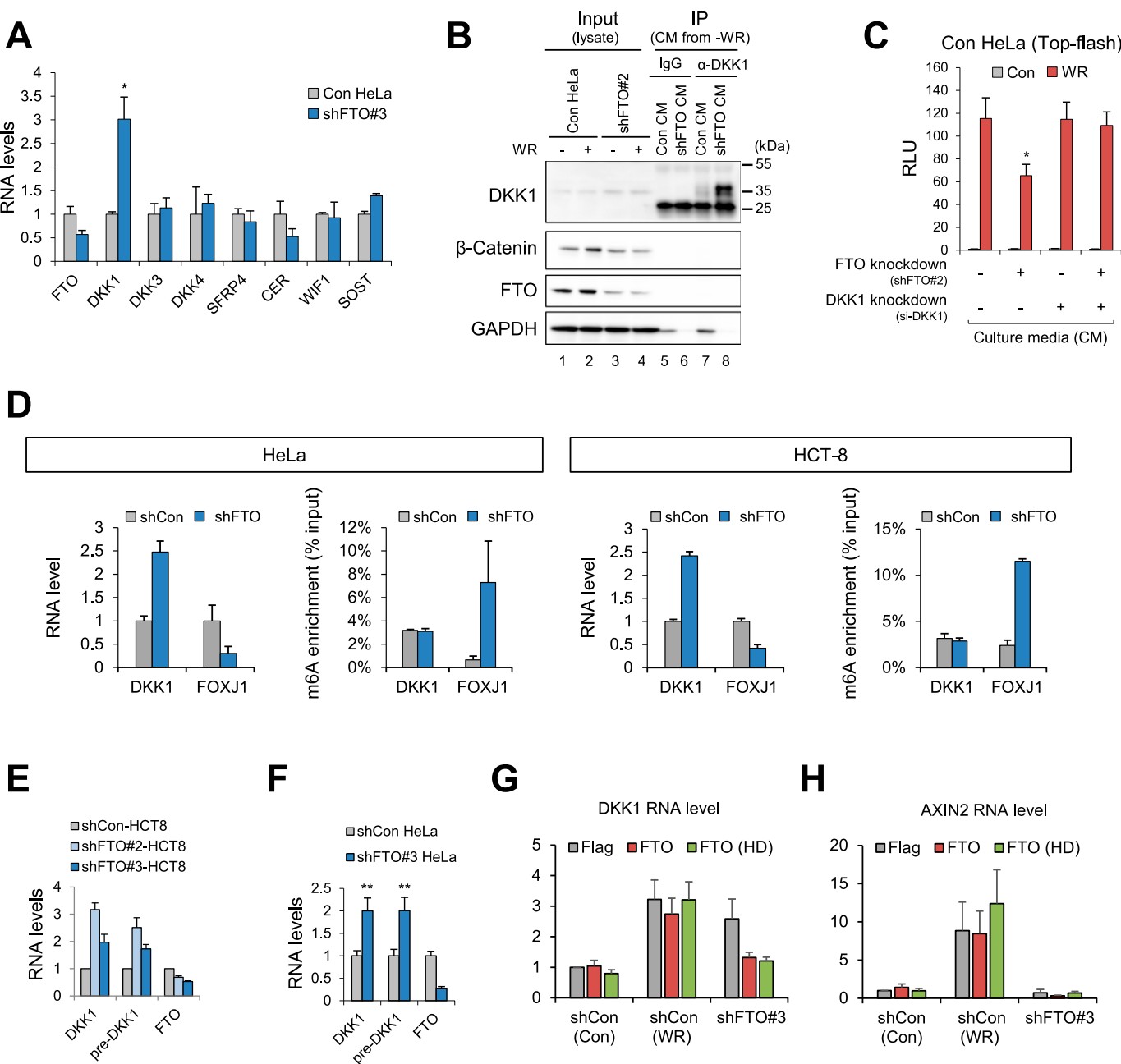

**Figure 3. Loss of FTO increases DKK1 expression.**
**(A)** RNA levels of various extracellular WNT antagonists were measured by RT-qPCR from control or FTO-depleted HeLa cells (shFTO#3). GAPDH was used as a normalization control (n = 3). **(B)** Secreted DKK1 protein levels were measured by Western blot after immunoprecipitating DKK1 from culture medium of control (Con CM) or FTO-depleted (shFTO#2) HeLa cells (shFTO CM). IgG immunoprecipitation serves as a negative control. **(C)** Culture medium of HeLa cells transiently expressing WNT reporter (TOP-flash) was replaced with the medium harvested from DKK1 siRNA or control siRNA (si-NC) transfected HeLa cells (control or FTO-depleted). Relative luciferase activities were measured after 16 h of WNT stimulation (WR, WNT3a, and R-Spondin1) (n = 6). **(D)** Unfragmented poly(A)+ RNAs from control or FTO-depleted HeLa or HCT-8 cells were immunoprecipitated with anti-m6A antibody. m6A enrichments were calculated as relative amounts of m6A immunoprecipitated fraction compared with input. Fold changes were compared after GAPDH normalization between input samples of control and FTO depletion (n = 3). **(E)** RNA levels of mature and pre-spliced form of DKK1 (pre-DKK1) were measured by RT-qPCR from shCon or two different shFTO (#2 and #3) expressing HCT-8 cells. GAPDH was used as a normalization control (n = 3). **(F)** RNA levels of mature and pre-spliced form of DKK1 were measured by RT-qPCR from control or FTO-depleted HeLa cells (shFTO#3). GAPDH was used as a normalization control (n = 3). **(G)** RNA levels of DKK1 were measured by RT-qPCR from control or FTO-depleted cells transfected with empty Flag vector, wild-type FTO or catalytically inactive FTO (FTO HD) treated with control or WR (n = 3). **(H)** RNA levels of AXIN2 were measured by RT-qPCR from control or FTO-depleted cells transfected with empty Flag vector, wild-type FTO or catalytically inactive FTO (FTO HD) treated with control or WR (n = 3). GAPDH was used as a loading control for Western blot. WR, WNT3A, and R-Spondin1. Error bars are ±SEM. One-sided t test, *P < 0.05, **P < 0.01.

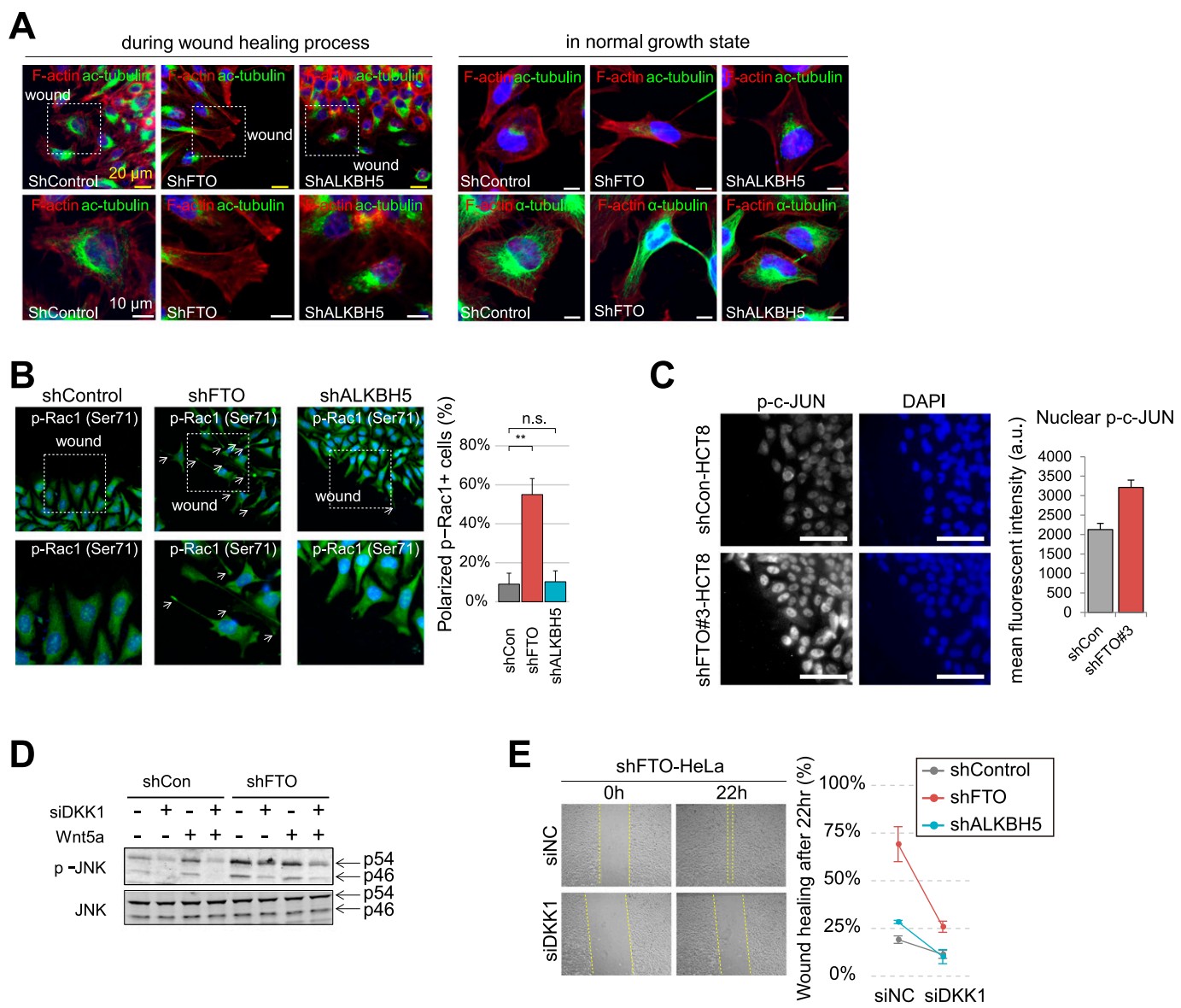

**Figure 4. Loss of FTO enhances cell migration by activating WNT/PCP signaling.**
**(A)** Filamentous actin (F-actin) and acetylated Tubulin (ac-tubulin) were visualized in cells during wound healing (6 h after scratch, see the Materials and Methods section) or in normal growth state. Regions enclosed by the dashed rectangles were enlarged and shown below. Directions of wound scratch were indicated. Scale bars are either 20 μm (yellow bars) or 10 μm (white bars). **(B)** Activated Rac1 was visualized using anti-p-Rac1 (Ser71) antibody in cells after 6 h of scratch wounding. Cells with polarized p-Rac1 staining near the wound edge were counted and graphed (right) (shCon, n = 3; shFTO, n = 4; shALKBH5, n = 3). Regions enclosed by the dashed rectangles were enlarged and shown below. Directions of wound scratch were indicated. **(C)** Phosphorylated c-JUN was visualized in HCT-8 cells after 6 h of scratch wounding near the wound edge (n = 5). Mean fluorescent intensities of nuclear p-c-JUN were measured with ImageJ software and graphed (right). Scale bars are 100 μm. **(D)** Levels p54 and p46 JNK and its activated form (p-JNK) were assessed by Western blot. Control or FTO-depleted HeLa cells were transiently transfected with either control siRNA (−) or DKK1 siRNA (+). Cells were treated with recombinant WNT5a protein for 12 h. **(E)** FTO-depleted HeLa cells along with control of ALKBH5-depleted cells were further transfected with either control (siNC) or DKK1 siRNA (siDKK1). Scratch wound-healing assays were performed 48 h after siRNA transfection. Representative wound healing of FTO-depleted cells were shown on the left. Yellow dashed lines indicate wound edges. Percentage of wound healing after 22 h were calculated as "100 − (wound width at 0 h − wound width at 22 h/wound width at 0 h)" (n = 3). Error bars are ± SEM. One-sided t test, **P < 0.01, n.s. P > 0.05.

the RNA levels of epithelial–mesenchymal transition markers (Fig S4E). Along with the increased cell motility and polarity, active Rac1 (i.e., Ser71-phosphorylated Rac1) was accumulated at the protruding membrane of FTO-silenced cells (Fig 4B). Moreover, phosphorylated c-JUN at the nucleus of FTO-silenced cells were also increased (Fig 4C), supporting the possibility of FTO-dependent

WNT/PCP regulation. ALKBH5-depleted cells did not exhibit change in cell polarity and motility (Figs 4A and B and S4A, C, and D).

The canonical WNT antagonist DKK1 is also known to activate the WNT/PCP signaling pathway in a number of biological contexts (Caneparo et al, 2007; Killick et al, 2014; Sellers et al, 2018). Thus, we hypothesized that the FTO-dependent up-regulation of DKK1

transcription (Figs 3 and S3) may also be responsible for the enhanced cell migration via WNT/PCP activation. To test this hypothesis, we treated FTO-depleted cells with WNT5A, a known WNT/PCP ligand (Ohkawara et al, 2011). Indeed, FTO knockdown augmented phosphorylation of JNK, c-JUN, and Dvl2 (Figs 4C and D and S4F and G), which are the hallmarks of WNT/PCP activation (Yamanaka et al, 2002). This increased JNK phosphorylation was abrogated after DKK1 knockdown using siRNA (Fig 4D), indicating that the up-regulation of DKK1 is responsible for the WNT/PCP activation in FTO-depleted cells. DKK1 knockdown also suppressed the enhanced cell migration of FTO-depleted cells (Fig 4E), demonstrating the role of FTO-dependent DKK1 up-regulation in cell migration via WNT/PCP activation. Of note, we cannot exclude the possibility of DDK1's other involvement in the activation of Rac1 that is independent of the WNT/PCP pathway.

## Discussion

In this study, we apply a combination of experimental and computational approaches to characterize the functional landscape of FTO in a data-driven manner. Comparative RNA-seq analysis of FTO- and ALKBH5-depleted cells reveal the functional association of FTO and WNT signaling pathways, specifically the canonical WNT/$\beta$-Catenin pathway and the non-canonical WNT/PCP pathway. We experimentally confirm the attenuation of the canonical WNT/$\beta$-Catenin pathway in FTO-depleted cells which is consistent with previous reports in zebra fish, mouse embryonic fibroblasts, and HEK293T cells (Osborn et al, 2014). Specifically, Osborn et al showed that loss of FTO causes developmental defects in zebra fish such as in neural crest cell lineages. However, the molecular mechanism of how FTO regulates WNT signaling was largely unknown.

Here, we discover that the WNT inhibition in FTO-depleted cells operates in a non-cell autonomous manner via DKK1. DKK1 is a known inhibitor of the canonical WNT/$\beta$-Catenin pathway (Glinka et al, 1998). This up-regulation of DKK1 also enhances the non-canonical WNT/PCP pathway which is consistent with our computational analysis. Contrary to FTO's well-known role in mRNA demethylation (Jia et al, 2011; Wei et al, 2018) and mRNA stability (Su et al, 2018; Kim et al, 2021), we find no significant change in *DKK1* mRNA methylation and stability in FTO-depleted cells. Instead, FTO regulates *DKK1* transcription in a m6A(m)-independent manner. Previous reports have suggested the role of FTO as a transcriptional regulator (Wu et al, 2010; Liu et al, 2019; Tao et al, 2020). This model aligns well with how other RNA-binding proteins interact with chromatin and regulates epigenetic and transcriptional states (Ji et al, 2013; Naftelberg et al, 2015). There are also reports of specific biological contexts where the canonical WNT pathway is activated after FTO depletion (Jeschke et al, 2021), further highlighting the importance of elucidating the exact molecular mechanism of FTO-dependent DKK1 regulation.

The multifacetedness of WNT signaling is often attributed to the opposing effects of WNT proteins (e.g., WNT3a and WNT5a), WNT receptors (e.g., FZD family of G protein couple receptors), co-receptors (e.g., LRP6 and ROR2), and its dual regulators of the canonical and non-canonical pathways. For example, Dishevelled

(DVL) proteins interact with GSK3 to inhibit $\beta$-Catenin protein degradation, but also play a role in the activation of JNK, AP-1, and other components of the noncanonical WNT pathway (Boutros et al, 1998; Li et al, 1999; Habas et al, 2003). Frat oncoproteins are reported to behave in a similar manner as DVL (van Amerongen et al, 2010). We extend this list of branch point regulators by investigating multiple WNT signaling pathways in FTO-depleted cells. In particular, we find that FTO regulates the expression of DKK1, another dual WNT regulator (Grumolato et al, 2010; Sellers et al, 2018), thereby contributing to the bifurcation of the canonical and noncanonical pathways. Of note, FTO is also associated with the noncanonical WNT/Ca$^{2+}$ pathway (Osborn et al, 2014). Although the exact molecular factor responsible for this association remains unknown, the report supports our observation of this intricate entanglement between FTO and multiple WNT signaling pathways. In all, this study proposes DKK1-mediated WNT signaling as a plausible explanation underlying the various functions of FTO in development and human diseases.

## Materials and Methods

### Plasmids, oligonucleotides, antibodies, and reagents

Wild-type Flag-hFTO and its catalytic mutant Flag-hFTO (HD) were gifts from Dr. Chuan He. Control shRNA in pLKO.puro backbone was a gift from Dr. David Sabatini (#1864; Addgene). pSpCas9(BB)-2A-GFP (PX458) was a gift from Feng Zhang (plasmid # 48138; Addgene; http://n2t.net/addgene:48138; RRID:Addgene_48138). ShRNA oligonucleotides against human FTO, METTL3, YTHDF2, and ALKBH5 were synthesized (Cosmogenetech) and ligated to pLKO1.puro (#8453; Addgene). Sequences of shRNA oligonucleotides are as follows:

shFTO#2 F:
5′-CCGGCGGTTCACAACCTCGGTTTAGCTCGAGCTAAACCGAGGTTGTG-AACCGTTTTTG-3′
shFTO#2 R:
5′-AATTCAAAAACGGTTCACAACCTCGGTTTAGCTCGAGCTAAACCGAG-GTTGTGAACCG-3′
shFTO#3 F
5′-CCGGCCCATTAGGTGCCCATATTTACTCGAGTAAATATGGGCACCTAA-TGGGTTTTTG-3′
shFTO#3 R
5′-AATTCAAAAACCCATTAGGTGCCCATATTTACTCGAGTAAATATGGGC-ACCTAATGGG-3′
shFTO#4 F
5′-CCGGACCTGAACACCAGGCTCTTTACTCGAGTAAAGAGCCTGGTGTT-CAGGTTTTTG-3′
shFTO#4 R
5′-AATTCAAAAACCTGAACACCAGGCTCTTTACTCGAGTAAAGAGCCTG-GTGTTCAGGT-3′
shALKBH5 F:
5′-CCGGCCTATGAGTCCTCGGAAGATTCTCGAGAATCTTCCGAGGACTC-ATAGGTTTTTG-3′
shALKBH5 R:
5′-AATTCAAAAACCTATGAGTCCTCGGAAGATTCTCGAGAATCTTCCGAG-GACTCATAGG-3′

shMETTL3#2 F
5′-CCGGGCCAAGGAACAATCCATTGTTCTCGAGAACAATGGATTGTTCCT-
TGGCTTTTTG-3′
shMETTL3#2 R
5′-AATTCAAAAAGCCAAGGAACAATCCATTGTTCTCGAGAACAATGGATT-
GTTCCTTGGC-3′
shYTHDF2#1 F
5′-CCGGGCTACTCTGAGGACGATATTCCTCGAGGAATATCGTCCTCAGA-
GTAGCTTTTTG-3′
shYTHDF2#1 R
5′-AATTCAAAAAGCTACTCTGAGGACGATATTCCTCGAGGAATATCGTCC-
TCAGAGTAGC-3′

siRNAs duplexes were synthesized (Bioneer) and used. Sequences of siRNAs are as follows:
si-DKK1: 5′-CCUGUCCUGAAAGAAGGUCAA-3′ (sense strand).
si-control: 5′-CCUACGCCACCAAUUUCGU-3′ (sense strand).

Primary antibodies used were anti-$\beta$-Catenin (#sc-7963; Santa Cruz), anti-acetylated Tubulin (#T7451; Sigma-Aldrich), anti-$\alpha$ Tubulin (#ab52866; Abcam), anti-FTO (#ab124892; Abcam), anti-GAPDH (#sc-32233 and #sc-25778; Santa Cruz), anti-m6A (#202003; SYSY), anti-RNA Polymerase 2 (#sc-56767; Santa Cruz), anti-DKK1 (#AF1096; R&D Systems and #sc-374574; Santa Cruz), anti-p-JNK (#4668; Cell Signaling), anti-JNK (#9258; Cell Signaling), anti-p-Rac1 (Ser71) (#2461; Cell Signaling), anti-Axin2 (#2151; Cell Signaling), anti-C-Myc (#5605; Cell Signaling), anti-p-c-Jun (#sc-822; Santa Cruz), and anti-Dvl2 (#3216; Cell Signaling).

Recombinant murine Wnt3a (#315-20; PeproTech), human/mouse Wnt5a (#645-WN-010; R&D Systems) and human R-Spondin1 (#4645-RS-025; R&D Systems) were purchased and used. GSK3 inhibitor SB216763 (#S3442; Sigma-Aldrich) was dissolved in DMSO and used at 10 $\mu$M. Alexa Fluor 647 Phalloidin (#A22287; Invitrogen) was used to stain F-actin.

## Cell culture, lentiviral shRNA delivery, and transfection

HeLa, 293T, HepG2, HCT-8, and SW480 cells were cultured in DMEM containing 10% FBS. Lentiviral supernatants were produced using Lenti-X Packaging Single Shots (VSV-G) (#631275; Clontech) mixed with pLKO.1 puro-based lentiviral constructs transfected to Lenti-X 293T cells. 24 h after transfection, supernatants were collected and concentrated using Lenti-X Concentrator (#631231; Clontech). Stable shRNA-expressing cell lines were established by infecting cells with shRNA-containing lentiviral supernatants together with 4 $\mu$g/ml protamine sulfate for 24 h. After 24–48 h of culture with fresh medium, cells were selected with medium containing 1~2 $\mu$g/ml puromycin (#P-1033; AG Scientific) until non-transduced cells became completely dead, which usually takes 2–3 d. Plasmid DNAs were transfected using Fugene HD (#E2312; Promega) according to the manufacturer's protocol.

## FTO and ALKBH5 knockout cell preparation

FTO or ALKBH5 knockout HeLa cells were prepared using CRISPR/Cas9 system as described previously (Kim et al, 2020). Briefly, HeLa cells were transfected with pSpCas9(BB)-2A-GFP-px458 plasmid (#48138; Addgene) with a previously described single-guide RNA against FTO (AGCTTCGCGCTCTCGTTCCT, PAM sequence is CGG) (Wei et al, 2021) or ALKBH5 (GGGACACCACCTCGTCAATG, PAM sequence is CGG). Clonal selection was performed using limiting dilution with an average plating density of 0.5 cells/well in the 96-well plate format and the mutation was initially tested using T7E1 (T7 Endonuclease I, #M0302; NEB) assay at 14 d post-seeding. Frame-shift mutations were confirmed using Sanger sequencing and the changed genomic sequences are listed in Table 1, with inserted and deleted sequences marked in red and dashes, respectively.

## WNT reporter assays

Luciferase activities were measured using Dual-Luciferase assay system (#E1980; Promega) according to the manufacturer's protocol. WNT reporter plasmid (TOP-flash) or control reporter (FOP-flash) together with CMV-Renilla plasmid were transiently transfected to shRNA-expressing cells. For co-culture experiments, reporter plasmids were transfected to either control or shFTO-expressing HeLa cells. After 24 h of transfection, transfected cells were co-cultured with shRNA-expressing cells with 1:1 ratio. Mixed cells were harvested after additional 24 h and luciferase activities were measured. For CM (culture medium or conditioned medium) experiments, culture media were harvested from shRNA-expressing cell cultures at least after 24 h of media change. Collected media were then filtered using syringe filter (#SLGP033RS; Millipore) before applying to the WNT reporter-transfected cell cultures. For immunodepletion of DKK1 protein, collected and filtered culture medium was incubated with anti-DKK1 antibody or with normal IgG

**Table 1. FTO and ALKBH5 KO Sanger sequencing.**

| sgRNAs for FTO or ALKBH5 knockouts | |
|---|---|
| sgRNA for FTO | AGCUUCGCGCUCUCGUUCCU |
| sgRNA for ALKBH5 | GGGACACCACCUCGUCAAUG |
| Genomic sequences of FTO and ALKBH5 knockout HeLa cells | |
| Parental (Chr16 reverse: 53704239-53704190) | cccgacataccttagcttcgcgctctcgttcctcggcagtcggggtgcgc |
| FTO KO allele 1 | cccgacataccttagcttcgcgctctcgttcctcggca——gggtgcgc |
| FTO KO allele 2 | cccgacataccttagcttcgcgctctcgttcctcggcaCgtcggggtgcg |
| Parental (Chr17 reverse: 18184615-18184556) | accacctcgtcaatgcgggcctcgatcttggcgcactcgtcctggctgaagaggcgcatc |
| ALKBH5 KO allele 1 | accacctc——————gggcctcgatcttggcgcactcgtcctggctgaagaggcgcatc |
| ALKBH5 KO allele 2 | accacctcgtc-atgcgggcctcgatcttggcgcactcgtcctggctgaagaggcgcatc |

overnight at 4°C. Antibody-protein complex was then removed with excessive protein A/G agarose bead.

### RNA isolation, quantitative RT-PCR

Total RNAs were extracted with TRIzol (Invitrogen) and treated with DNase I (Takara). 200–500 ng of total RNAs were reverse transcribed using Primescript RT master mix (Takara). Quantitative PCR was performed using Power SYBR Green PCR Master Mix (Thermo Fisher Scientific) under StepOnePlus Real-Time PCR system (Thermo Fisher Scientific). Primer sequences are listed in Table 2.

### Western blot and ubiquitination assay

For Western blots, cells were lysed with a lysis buffer containing 20 mM Hepes, pH 7.2, 150 mM NaCl, 1% Triton X-100, 10 mM EDTA, pH 8.0, and 1 mM DTT, together with protease inhibitor cocktail (Promega) and phosphatase inhibitor cocktail (Promega). Cell lysates were separated on Novex WedgeWell Tris-Glycine Mini Gels (Thermo Fisher Scientific) and transferred to polyvinylidene fluoride membrane (GE Healthcare) or to nitrocellulose membrane (Amersham). Blots were washed with PBST (PBS containing 0.1% Tween-20), blocked with 3% BSA for 1 h at room temperature and incubated with primary antibodies overnight at 4°C. Membranes were then washed three times with PBST and incubated with secondary antibodies for 1 h at room temperature. Secondary antibodies were from Li-Cor (IRDye 680 and IRDye 800, 1:10,000) for

infrared imaging of proteins using the LiCor Odyssey system. For chemiluminiscence detection of signals, HRP-conjugated secondary antibodies (Jackson Laboratories) were used.

Ubiquitination assays were performed as described (Kim et al, 2015). Briefly, HeLa cells were transfected with Flag-FTO together with HA-ubiquitin. After 48 h of transfection, cells were treated either with DMSO or with SB-216763, together with a proteasomal inhibitor MG132 for 2–4 h. Cells were then lysed with lysis buffer containing 20 mM Hepes, pH 7.2, 150 mM NaCl, 1% Triton X-100, 10 mM EDTA, pH 8.0, 1 mM DTT, and 1 mM NaF together with a protease inhibitor cocktail and incubated in Eppendorf tubes at 95°C in the presence of 1% SDS to disrupt protein–protein interactions. The lysates were then diluted 10-fold to reduce SDS concentration to 0.1% before immunoprecipitating with Flag antibody conjugated agarose beads (#A2220; Sigma-Aldrich). Bead-lysate mixtures were incubated overnight at 4°C using an end-over-end rotator. Beads were then washed three times with cold buffer (10 mM Tris, pH 9.0, 100 mM NaCl, and 0.5% NP40) followed by elution with SDS loading buffer at 95°C for 5 min.

### DKK1 secretion assay

Secreted DKK1 levels were measured either by Western blot after immunoprecipitating DKK1 from the culture medium using anti-DKK1 (#AF1096; R&D Systems) or by commercially available ELISA assay (Human Dkk-1 Quantikine ELISA Kit, #DKK100B; R&D Systems) according to the manufacturer's protocol. For DKK1 ELISA,

**Table 2. qPCR primers used in the study.**

| Human RT-qPCR | Forward | Reverse |
|---|---|---|
| GAPDH | TGGGTGTGAACCATGAGAAG | GGGTGCTAAGCAGTTGGTG |
| FTO | ACTTGGCTCCCTTATCTGACC | TGTGCAGTGTGAGAAAGGCTT |
| DKK1 | CCTTGAACTCGGTTCTCAATTCC | CAATGGTCTGGTACTTATTCCCG |
| DKK3 | AGGACACGCAGCACAAATTG | CCAGTCTGGTTGTTGGTTATCTT |
| DKK4 | ACGGACTGCAATACCAGAAAG | CGTTCACACAGAGTGTCCCAG |
| SFRP4 | CCTGGAACATCACGCGGAT | CGGCTTGATAGGGTCGTGC |
| CER | GGATGGCCGCCAGAATCAG | TGGCACTGCGACAAACAGAT |
| WIF1 | TCTCCAAACACCTCAAAATGCT | GACACTCGCAGATGCGTCT |
| SOST | ACACAGCCTTCCGTGTAGTG | GGTTCATGGTCTTGTTGTTCTCC |
| AXIN2 | CAACACCAGGCGGAACGAA | GCCCAATAAGGAGTGTAAGGACT |
| TCF3 | TGCAGTGAGCGTGAAATCACCAGT | AATGGCTGCACTTTCCTTCAGGGT |
| pre-DKK1 | CGGGCGGGAATAAGTACCAG | CAGATAGGACCCTTTCAAGGTCAG |
| ZO-1 | CAACATACAGTGACGCTTCACA | CACTATTGACGTTTCCCCACTC |
| MMP9 | TGTACCGCTATGGTTACACTCG | GGCAGGGACAGTTGCTTCT |
| N-CADHERIN | TCAGGCGTCTGTAGAGGCTT | ATGCACATCCTTCGATAAGACTG |
| SNAIL | TCGGAAGCCTAACTACAGCGA | AGATGAGCATTGGCAGCGAG |
| CTNNB1 | CATGTACGTTGCTATCCAGGC | CTCCTTAATGTCACGCACGAT |
| ALKBH5 | GCCTATTCGGGTGTCGGAAC | CTGAGGCCGTATGCAGTGAG |
| Human CHIP-qPCR | Forward | Reverse |
| DKK1 TSS | GCATAAAGGAGAGGGGCAAA | TACTTTACAGAGCCGAGGGG |
| HSP70 TSS | AGCCTCATCGAGCTCGGTGATTG | AAGGTAGTGGACTGTCGCAGCAGC |

absorbance was measured at 450 nm with a correction at 540 nm using the Eon microplate spectrophotometer (Agilent).

## Chromatin immunoprecipitation

For chromatin immunoprecipitation with RNA polymerase 2 antibody, cells were first fixed and cross-linked with 1% formaldehyde for 10 min and quenched with 0.25 M glycine before lysed with RIPA lysis buffer (50 mM Tris–HCl pH 7.5, 150 mM KCl, 5 mM EDTA, 1% NP40, 0.1% SDS, and 0.5% sodium deoxycholate supplemented with cocktails of protease inhibitor and phosphatase inhibitor) and sonicated to obtain DNA fragments around 500 bp. These samples were then precleared with pre-saturated protein A bead with yeast tRNA and salmon sperm DNA followed by the incubation with antibody-bead mixture overnight at 4°C. After extensive washing with RIPA wash buffer (50 mM Tris–HCl, pH 7.5, 150 mM KCl, 1% NP40, and 0.25% sodium deoxycholate) and TE buffer (10 mM Tris–HCl, pH 7.5, 1 mM EDTA), samples were resuspended in RIPA elution buffer (50 mM Tris–HCl, pH 7.5, 5 mM EDTA, 10 mM DTT, and 1% SDS) and reverse cross-linked overnight at 65°C. DNA fragments were purified using QIAquick PCR purification kit (QIAGEN) after proteinase K digestion for 1 h at 45°C. RNA polymerase 2 bound DNA fragments were quantified by quantitative PCR using primer sets listed in Table 2.

## Scratch wound healing assay

For scratch wound healing assay, cells were grown near confluency before making scratch wound using 1–200 $\mu$l micropipette tips. For positional reference, bottom side of culture plate was marked. Wound healing activities were traced and photographed under stereomicroscope. For immunocytochemistry near wound edge, cells were grown on the coverslip near confluency and scratch wound was introduced as described above.

## Immunocytochemistry

For immunocytochemistry, cells were grown on top of coverslip and fixed with 4% formaldehyde for 1 h at room temperature. Fixed cells were washed and permeabilized with PBST and blocked with 3% BSA for 1 h at room temperature. Primary antibodies were applied overnight at 4°C. After 1 h of secondary antibody incubation and extensive PBS washing, coverslips were mounted on the slide glass with Prolong Gold antifade reagent with DAPI (#P-36931; Life Technologies) and visualized under Zeiss LSM 700 confocal microscope.

## Sequence data analysis

Pre-processed siFTO and siALKBH5 knockdown RNA-seq data were downloaded from Gene Expression Omnibus: GSE78040 (Mauer et al, 2017). Pre-process shFTO knockdown RNA-seq data on HepG2 and K562 cells were downloaded from the ENCODE portal (ENCODE Project Consortium, 2012; Davis et al, 2018) (https://www.encodeproject.org/) the following identifiers: ENCFF019JYU, ENCFF375TPO, ENCFF605EOA, ENCFF622ZUI, ENCFF041CFW, ENCFF739AIP, ENCFF826FQP, and ENCFF930LYD. BaseMean, regularized fold-change, and differential gene expression analysis were conducted using DESeq2 v1.14.1 (Love et al, 2014) for genes with more than 1 aligned read.

Transcriptional targets of LEF1, TCF7, ATF2, and ATF4 were from TFactS (Essaghir et al, 2010) and download on 6 March 2019. Canonical WNT targets were downloaded from the WNT homepage (https://web.stanford.edu/group/nusselab/cgi-bin/wnt/target_genes) and downloaded on 21 Nov 2018.

## Gene set enrichment analysis

Gene set enrichment scores were computed using the PAGE algorithm (Kim & Volsky, 2005). Briefly, the Central Limit Theorem states that the distribution of the average of randomly sampled n observations approaches the normal distribution for larger n. PAGE uses this theorem to model the null distribution and estimate the effect size (i.e., Z score) and P-value of a given set of genes. Specifically, given the regularized fold change of gene $g_i$ and a gene set $S = (g_1, g_2, ..., g_n)$, the gene set enrichment score $z_s$ is the following:

$$z_s = \frac{m_s - \mu}{\sigma / \sqrt{n}},$$

where $m_s$ is the sample mean, $\mu$ is the population mean, and $\sigma$ is the population variance. 487 WikiPathway gene sets (Kutmon et al, 2016) were downloaded on 7 March 2019. Mouse Gene Ontology terms and gene sets (Ashburner et al, 2000; Carbon et al, 2009; The Gene Ontology Consortium, 2019) were downloaded on 5 April 2016. Only gene sets of at least size 10 were used for further analysis.

# Data Availability

The authors will comply with Life Science Alliance policies for the sharing of research materials and data.

# Supplementary Information

# Acknowledgements

We thank Chuan He for FTO plasmids; Jeongwoon Lee for assistance in cell migration assays; Da-Eun Choi, Eunji Kim, and Jihye Yang for technical support and help in plasmids construction; and Narry Kim for discussion and general advice. We thank the ENCODE Consortium and the Brenton Graveley lab for generating the shFTO RNA-seq datasets. We also thank members of Narry Kim's laboratory for discussions and comments. This work was supported by IBS-R008-D1 of the Institute for Basic Science from the Ministry of Science and ICT of Republic of Korea. This research was also supported by the Basic Science Research Program through the National Research Foundation of Korea (NRF) funded by the Ministry of Education (NRF-2021R1C1C1009282 to Y Lee).

## Author Contributions

H Kim: conceptualization, data curation, formal analysis, supervision, validation, investigation, visualization, methodology, and writing—original draft, review, and editing.

S Jang: validation, investigation, and visualization.

Y Lee: conceptualization, data curation, software, formal analysis, supervision, investigation, visualization, methodology, and writing—original draft, review, and editing.

## Conflict of Interest Statement

The authors declare that they have no conflict of interest.

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
