## [Reviewer comments · Life Science Alliance]

Life Science Alliance

The m6A(m)-independent role of FTO in regulating WNT signaling pathways

Hyunjoon Kim, Soohyun Jang, and Young-suk Lee

DOI: <https://doi.org/10.26508/lsa.202101250>

Corresponding author(s): Hyunjoon Kim, Institute for Basic Science and Young-suk Lee, Korea Advanced Institute of Science and Technology

Review Timeline:

Submission Date:	2021-10-03
Editorial Decision:	2021-10-26
Revision Received:	2022-01-24
Editorial Decision:	2022-01-31
Revision Received:	2022-01-31
Accepted:	2022-02-01

Scientific Editor: Novella Guidi

Transaction Report:

October 26, 2021

Re: Life Science Alliance manuscript #LSA-2021-01250

Dr. Hyunjoon Kim
Institute for Basic Science
Center for RNA Research, IBS
Bldg 203-521, Seoul National University
1 Gwanangno, Gwanak-gu
Seoul 08826
Korea, Republic of (South Korea)

Dear Dr. Kim,

Thank you for submitting your manuscript entitled "The m6A(m)-independent role of FTO in regulating WNT signaling pathways" to Life Science Alliance. The manuscript was assessed by expert reviewers, whose comments are appended to this letter. As you will note from the reviewers' comments below, all the reviewers are enthusiastic about the work but do raise some important concerns that need to be addressed before the paper can be considered further. Rev1 main concerns are that data do not show a strong correlation between the FTO expression levels, TOP-Flash activity and the DKK1 transcription levels and suggests performing parallel experiments by knocking out FTO and then examine the DKK1 levels. Also, to be consistent with Figure 1, the TOP-Flash activity and β -Catenin expression level should be tested in ALKBH5 knockdown or knockout cell lines. This reviewer also requests to determine whether WNT3a or DKK1 is more important for regulating WNT signaling after FTO knockdown by evaluating the absolute expression of WNT and DKK1 protein in the culture medium after FTO knockdown or knockout. Rev2 would like to see some in vivo data that FTO regulates canonical and non-canonical pathway to further confirm the physiological significance of FTO involved in Wnt signaling pathway. Rev3's concern is the lack of molecular insights of FTO-dependent DKK1 to be addressed with additional experiments or text discussion. Also, there is a lack of experimental prove that noncanonical WNT/PCP pathway is activated to be address by either additional experiments or by toning down statements. We, thus, encourage you to submit a revised version of the manuscript back to LSA that responds to all of the reviewers' points.

Thank you for this interesting contribution to Life Science Alliance. We are looking forward to receiving your revised manuscript.

Sincerely,

B. MANUSCRIPT ORGANIZATION AND FORMATTING:

Reviewer #1 (Comments to the Authors (Required)):

In this manuscript, the authors described an association between FTO and WNT signaling pathways. They reported that FTO depletion causes attenuation of canonical WNT/ β -Catenin signal and activation of non-canonical WNT pathway by upregulating DKK1 mRNA. And FTO regulates DKK1 in a manner independent of FTO demethylase activity. The authors showed that culture medium from FTO knockdown cells can attenuate WNT3a induced TOP-Flash activity and β -Catenin accumulation, and this attenuated TOP-Flash activity can be rescued by DKK1 siRNA or anti-DKK1 antibody. This is a piece of interesting work. But the manuscript requires significant improvement before can be considered.

1. In Figure 2A, the authors showed 3 different shRNA (2#,3# and 4#) knockdown FTO in HeLa cells, and the FTO expression is shown in Figure 2B, Figure 3B and Figure S3B. It is somehow strange that the authors' data do not show a strong correlation between the FTO expression levels and TOP-Flash activity and the DKK1 transcription levels. I would suggest the authors to perform parallel experiments by knockout FTO and then examine the DKK1 levels upon FTO knockout.
2. In Figure 1, the authors used the ALKBH5 siRNA knockdown RNA-Seq dataset as a negative control. To be consistent with Figure 1, the TOP-Flash activity and β -Catenin expression level should be tested in ALKBH5 knockdown or knockout cell lines.
3. Previously, Osborn et al. (2014) have described that FTO-deficient MEFs and 293T cells showed down-regulation of WNT signaling by decreasing the stability of β -Catenin and abrogating the translocation of β -Catenin in to the nucleus. In this manuscript, the authors' data also showed less β -Catenin level in FTO knockdown cells. The importance of DKK1 in FTO-mediated down-regulation of WNT activity should be further explained.
4. The authors showed that DKK1 mRNA only increased 2-3 times in FTO knockdown cells, but the secreted DKK1 protein was substantially increased in the culture medium of FTO knockdown cells. If DKK1 competes with WNT3a to inhibit WNT signaling activation, the authors should evaluate the absolute expression value of WNT and DKK1 protein in the culture medium after FTO knockdown or knockout. This experiment can determine whether WNT3a or DKK1 is more important for regulating WNT signaling in a non-cell-autonomous manner after FTO knockdown.

Reviewer #2 (Comments to the Authors (Required)):

In this report, the authors explore the role of FTO demethylase in the regulation of Wnt pathway. Through re-analysis of public database and several cell-based experiment the authors suggest functional interaction between FTO and Wnt signaling pathway. Loss of FTO repressed the canonical Wnt/ β -catenin pathway whereas it upregulated noncanonical pathway by regulating DKK1 transcription, but not mRNA methylation. Although the molecular mechanism is still not clearly elucidated in this manuscript, this regulatory function independent of demethylase activity is of high interest to me and will expand new approaches in the study of FTO-related biological processes. Overall, the presented data are of good quality and the paper is well written and easy to follow. However, some points have not been addressed sufficiently and, therefore, the manuscript requires amendments to warrant publication.

Below are some points to be addressed by the authors to strengthen or extend their observations.

Major points,

1. To further confirm the physiological significance of FTO involved in Wnt signaling pathway, the authors should show in vivo data that FTO regulates canonical and non-canonical pathway.

2. In Figure 2B, it is likely that WR induces FTO abundance as well as β -catenin. Similar results were observed in other data including supplementary figure 2A, 2C and 3H. I would suggest that the author should test it to be clear and include it in the revised manuscript.

Minor points,

1. HepG2 carry two different β -catenin (CTNNB1) alleles: one encoding wild-type β -catenin and the other encoding a constitutively active β -catenin bearing a exon 3-4 deletion in the N terminus encompassing the GSK3 phosphorylation sites as mentioned in the manuscript. In Supplementary Figure S2E, the authors use an antibody recognizing C-terminal region, but β -catenin was observed only in truncated size.

Reviewer #3 (Comments to the Authors (Required)):

Kim et al. used computational approaches to characterize the functional landscape of FTO depletion and got clues functional interactions between FTO and Wnt signaling pathways. Authors found that canonical Wnt/ β -catenin signaling is inhibited in FTO depleted cells by enhancing the level of DKK1 and the upregulation of DKK1 promotes cell migration via activating the noncanonical WNT/PCP pathway. Although they did not perform mechanical studies they showed that the mRNA demethylation activity of FTO is not necessary for the regulation of DKK1 level. I think the manuscript is well-written and the quality of data is clean enough to support their claims. This manuscript can be published if following concerns are resolved.

1. Although the authors admitted that further studies are necessary, if possible further discussion or additional experiments for the molecular mechanism of FTO-dependent DKK1 regulation will be required for the integrity of this manuscript.

2. Authors examined the shFTO cells' migration, morphology, and the levels of pRac1 and p-c-Jun, which led them claim that the noncanonical WNT/PCP pathway is activated in these cells. I think that these changes are characteristics of noncanonical WNT/PCP pathway, but authors cannot claim that noncanonical WNT/PCP pathway is activated in shFTO cells, because these phenomena can be caused simply by activation of Rac1. Authors need to show changes of noncanonical WNT/PCP pathway specific Wnts expression and Dvl phosphorylation in shFTO cells. If they cannot provide these data, authors need to tone-down their claim.

Response to Reviewer

We thank the reviewers for carefully reading our manuscript and providing their valuable feedback. We are grateful that all three reviewers agreed on the merit of our study on the demethylase-independent roles of FTO in regulating DKK1 expression and WNT signaling pathways. We now performed additional experiments and revised the text discussion to address the questions raised by the reviewers. In particular, we generated FTO and ALKBH5 knockout HeLa cell lines using CRISPR/CAS9 system as orthogonal approaches to support our previous shRNA-mediated knockdown results. We hope the reviewers find our responses clear and to the point.

Reviewer #1 (Comments to the Authors (Required)):

In this manuscript, the authors described an association between FTO and WNT signaling pathways. They reported that FTO depletion causes attenuation of canonical WNT/ β -Catenin signal and activation of non-canonical WNT pathway by upregulating DDK1 mRNA. And FTO regulates DKK1 in a manner independent of FTO demethylase activity. The authors showed that culture medium from FTO knockdown cells can attenuate WNT3a induced TOP-Flash activity and β -Catenin accumulation, and this attenuated TOP-Flash activity can be rescued by DKK1 siRNA or anti-DKK1 antibody. This is a piece of interesting work. But the manuscript requires significant improvement before can be considered.

We are grateful that the reviewer finds our study interesting. Adhering to the reviewer's suggestion, we now performed parallel experiments in FTO or ALKBH5 knockout cells.

1. In Figure 2A, the authors showed 3 different shRNA (2#,3# and 4#) knockdown FTO in HeLa cells, and the FTO expression is shown in Figure 2B, Figure 3B and Figure S3B. It is somehow strange that the authors' data do not show a strong correlation between the FTO expression levels and TOP-Flash activity and the DKK1 transcription levels. I would suggest the authors to perform parallel experiments by knockout FTO and then examine the DKK1 levels upon FTO knockout.

Adhering to the reviewer's suggestion, we generated FTO knockout HeLa cells to conduct parallel experiments regarding the expression of DKK1. DKK1 mRNA and pre-mRNA levels were upregulated in FTO KO cells (Fig. R1.1), which is consistent with our original shRNA experiments.

Fig. R1.1. DKK1 mRNA and pre-mRNA transcripts are up-regulated in FTO knockout (KO) cells. FTO KO HeLa cells were generated using CRISPR-CAS9 system. RNA levels were measured by RT-qPCR and normalized to GAPDH (n = 3).

2. In Figure 1, the authors used the ALKBH5 siRNA knockdown RNA-Seq dataset as a negative control. To be consistent with Figure 1, the TOP-Flash activity and β -Catenin expression level should be tested in ALKBH5 knockdown or knockout cell lines.

We generated ALKBH5 knockout HeLa cells and conducted parallel experiments to be consistent with our previous Figure 1. The DKK1 mRNA levels (Fig. R1.2; manuscript Fig. S3B), TOP-Flash activity (Fig. R1.3, left; manuscript Fig. S2A), β -Catenin protein levels (Fig. R1.3, right; manuscript Fig. S2B) were measured in both ALKBH5 KO and FTO KO cells. Consistent with our RNA-Seq analysis in Figure 1, ALKBH5 KO cells did not show increased DKK1 or pre-DKK1 RNA levels (Fig. R1.2) nor did it demonstrate comparable attenuation of canonical WNT activity (Fig. R1.3).

Fig. R1.2. DKK1 mRNA and pre-mRNA transcripts are up-regulated in FTO knockout (KO) cells but not ALKBH5 KO cells. RNA levels were measured by RT-qPCR and normalized to GAPDH (n = 3).

Fig. R1.3. WNT/ β -Catenin signaling is attenuated upon FTO knockout but not by ALKBH5 knockout. (A) FTO KO and ALKBH5 KO HeLa cells were transiently transfected with WNT reporter (TOP-Flash) and stimulated with WR (WNT3a + RSP01) for 12 hours. Relative luciferase activities were measured (n = 6). (B) Protein levels of β -CATENIN and FTO were measured by western blot. GAPDH was used as a loading control.

3. Previously, Osborn et al. (2014) have described that FTO-deficient MEFs and 293T cells showed down-regulation of WNT signaling by decreasing the stability of β -Catenin and abrogating the translocation of β -Catenin in to the nucleus. In this manuscript, the authors' data also showed less β -Catenin level in FTO knockdown cells. The importance of DKK1 in FTO-mediated down-regulation of WNT activity should be further explained.

We thank the reviewer for the suggestion. As mentioned by the reviewer, Osborn et al. (2014) reported the down-regulation of canonical WNT/ β -Catenin signaling in MEF and 293T cells and how this attenuation may inhibit adipogenic lineage commitment and disrupt proper embryonic development. Our work provides further mechanistic insight for this FTO-dependent WNT regulation. This point is now further explained in the discussion section.

4. The authors showed that DKK1 mRNA only increased 2-3 times in FTO knockdown cells, but the secreted DKK1 protein was substantially increased in the culture medium of FTO knockdown cells. If DKK1 competes with WNT3a to inhibit WNT signaling activation, the authors should evaluate the absolute expression value of WNT and DKK1 protein in the culture medium after FTO knockdown or knockout. This experiment can determine whether WNT3a or DKK1 is more important for regulating WNT signaling in a non-cell-autonomous manner after FTO knockdown.

To address the reviewer's concern, we measured the amount of secreted DKK1 proteins in FTO KO and ALKBH5 KO cells using the commercial DKK1 ELISA kit. Accumulation of secreted DKK1 proteins were only found in FTO KO cells (Fig. R1.4; manuscript Fig. S3E) which is consistent with our findings. As cited in this manuscript, it has been previously reported that DKK1 proteins act as WNT-antagonists (Glinka et al., 1998). In this study, we also presented two independent experimental evidences (manuscript Fig. 3C and S3F-G) that the canonical WNT signaling is attenuated in a DKK1-dependent manner. We hope the reviewer agrees that the exact protein dynamics of the stoichiometry of DKK1-dependent WNT regulation is beyond the scope of this study.

Fig. R1.4. Secreted DKK1 levels were measured from the culture medium of HeLa WT, FTO KO and ALKBH5 KO cells.

Reviewer #2 (Comments to the Authors (Required)):

In this report, the authors explore the role of FTO demethylase in the regulation of Wnt pathway. Through re-analysis of public database and several cell-based experiment the authors suggest functional interaction between FTO and Wnt signaling pathway. Loss of FTO repressed the canonical Wnt/ β -catenin pathway whereas it upregulated noncanonical pathway by regulating DKK1 transcription, but not mRNA methylation. Although the molecular mechanism is still not clearly elucidated in this manuscript, this regulatory function independent of demethylase activity is of high interest to me and will expand new approaches in the study of FTO-related biological processes. Overall, the presented data are of good quality and the paper is well written and easy to follow. However, some points have not been addressed sufficiently and, therefore, the manuscript requires amendments to warrant publication.

We are grateful that the reviewer agrees that this work will expand the study of FTO-related biological processes.

Below are some points to be addressed by the authors to strengthen or extend their observations.

Major points,

1. To further confirm the physiological significance of FTO involved in Wnt signaling pathway, the authors should show *in vivo* data that FTO regulates canonical and non-canonical pathway.

The physiological significance of FTO-dependent WNT regulation has been previously reported in Osborn et al. (2014). In this study, we focused on the molecular mechanism of this regulation in comparison to another m6A demethylase ALKBH5. We hope the reviewer agrees that it'd be sufficient to discuss the physiological significance of FTO-related biological processes in the discussion section and that further *in vivo* demonstration is beyond the scope of this work.

2. In Figure 2B, it is likely that WR induces FTO abundance as well as β -catenin. Similar results were observed in other data including supplementary figure 2A, 2C and 3H. I would suggest that the author should test it to be clear and include it in the revised manuscript.

We thank the reviewer for the suggestion. As mentioned by the reviewer, our previous experiments (manuscript Fig. 2B, S2C, S2E, and S3J) suggest that FTO proteins increase after WNT3a treatment. Canonical WNT signaling leads to the cytoplasmic stabilization of many other GSK3-target proteins besides β -Catenin, termed the WNT/STOP pathway (Acebron et al., 2014; Kim et al., 2015; Ploper et al., 2015). We find that the treatment of WNT3a and SB216763, a GSK3 inhibitor, increases the FTO protein levels (Fig. R2.1A;

manuscript Fig. S2I) but not the FTO mRNA level (Fig. R2.1B; manuscript Fig. S2H), indicating that FTO protein levels are regulated in a post-translational manner. Recently, it has been reported that GSK3 phosphorylates FTO to generate phosphodegron for polyubiquitination and proteasomal degradation (Faulds et al., 2018). Treatment of SB216763 diminishes the polyubiquitination of FTO (Fig. R2.2; manuscript Fig. S2J), suggesting FTO protein stabilization by GSK3 inhibition. Altogether this suggests FTO proteins as another target of the WNT/STOP pathway and the proteomic remodeling by canonical WNT signaling.

Fig. R2.1. WNT/STOP signaling stabilizes FTO protein. (A) Both WNT stimulation (WNT+RSPO) and GSK3 inhibition (SB216763) induce FTO protein stabilization along with β -Catenin stabilization. GAPDH was used as loading control. (B) FTO mRNA levels are unaffected by either WNT stimulation (WR) or GSK3 inhibition (SB216763). AXIN2 was used as positive control for WNT activity.

Fig. R2.2. GSK3 inhibition diminishes polyubiquitination of FTO. Polyubiquitination of Flag-FTO in the presence of proteasomal inhibitor (MG132) and GSK3 inhibitor (SB216763). HeLa cells were transiently transfected with either mock, HA-ub alone or Flag-FTO together with HA-ub. After Flag immunoprecipitation, ubiquitinated FTO were assessed by western blot using HA antibody. GAPDH was used as loading control.

Minor points,

1. HepG2 carry two different β -catenin (CTNNB1) alleles: one encoding wild-type β -catenin and the other encoding a constitutively active β -catenin bearing a exon 3-4 deletion in the N terminus encompassing the GSK3 phosphorylation sites as mentioned in the manuscript. In Supplementary Figure S2E, the authors use an antibody recognizing C-terminal region, but β -catenin was observed only in truncated size.

We apologize for the missing band of wild-type β -Catenin, which we have omitted from the original manuscript. We now include the wild-type β -Catenin band (Fig. R2.3; manuscript Fig. S2G).

A**B**
Fig. R2.3. Two forms of β -CATENIN are detected in HepG2 cells. (A) Both the wild-type and exon 3-4 deletion forms of β -Catenin are detected in HepG2 cells. (B) uncropped gel images.

Reviewer #3 (Comments to the Authors (Required)):

Kim et al. used computational approaches to characterize the functional landscape of FTO depletion and got clues functional interactions between FTO and Wnt signaling pathways. Authors found that canonical Wnt/b-catenin signaling is inhibited in FTO depleted cells by enhancing the level of DKK1 and the upregulation of DKK1 promotes cell migration via activating the noncanonical WNT/PCP pathway. Although they did not perform mechanical studies they showed that the mRNA demethylation activity of FTO is not necessary for the regulation of DKK1 level. I think the manuscript is well-written and the quality of data is clean enough to support their claims. This manuscript can be published if following concerns are resolved.

We appreciate the reviewer's positive evaluation on the quality of the manuscript.

1. Although the authors admitted that further studies are necessary, if possible further discussion or additional experiments for the molecular mechanism of FTO-dependent DKK1 regulation will be required for the integrity of this manuscript.

Adhering to the reviewer's suggestion, we now provide further discussion on the possible molecular mechanisms of FTO-dependent DKK1 regulation. Specifically, we amend the discussion section:

“ Instead, FTO regulates DKK1 transcription in a m6A(m)-independent manner. Previous reports have suggested the role of FTO as a transcriptional regulator (Liu et al., 2019; Tao et al., 2020; Wu et al., 2010). This model aligns well with how other RNA-binding proteins interact with chromatin and regulates epigenetic and transcriptional states (Ji et al., 2013; Naftelberg et al., 2015). There are also reports of specific biological contexts where the canonical WNT pathway is activated after FTO depletion (Jeschke et al., 2021), further highlighting the importance of elucidating the exact molecular mechanism of FTO-dependent DKK1 regulation.”

2. Authors examined the shFTO cells' migration, morphology, and the levels of pRac1 and p-c-Jun, which led them claim that the noncanonical WNT/PCP pathway is activated in these cells. I think that these changes are characteristics of noncanonical WNT/PCP pathway, but authors cannot claim that noncanonical WNT/PCP pathway is activated in shFTO cells, because these phenomena can be caused simply by activation of Rac1. Authors need to show changes of noncanonical WNT/PCP pathway specific Wnts expression and Dvl phosphorylation in shFTO cells. If they cannot provide these data, authors need to tone-down their claim.

Fig. R3.1. Dvl2 phosphorylation was increased in FTO depleted cells.

We thank the reviewer for closely reading our manuscript and providing an alternative explanation to our observations. As suggested by the reviewer, we include the reviewer's interpretation in the main text and thus tone down our claim (see text changes in red). However, we'd like to emphasize that the rescue experiments (manuscript Fig. 4D-E) by DKK1 depletion supports our interpretation as DKK1 has not been reported to directly regulate intracellular Rac1. Moreover, we do find a moderate increase in Dvl2 phosphorylation in FTO depleted cells (Fig. R3.1; manuscript Fig. S4G).

January 31, 2022

RE: Life Science Alliance Manuscript #LSA-2021-01250R

Dr. Hyunjoon Kim
Institute for Basic Science
Center for RNA Research, IBS
Bldg 203-521, Seoul National University
1 Gwanangno, Gwanak-gu
Seoul 08826
Korea, Republic of (South Korea)

Dear Dr. Kim,

Thank you for submitting your revised manuscript entitled "The m6A(m)-independent role of FTO in regulating WNT signaling pathways". We would be happy to publish your paper in Life Science Alliance pending final revisions necessary to meet our formatting guidelines.

- please add the Twitter handle of your host institute/organization as well as your own or/and one of the authors in our system
- please indicate scale bar size in Legend for figure 4A and add scale bars for figure 4C
- please provide a separate Data Availability section
- although the resolution of figure 4 is 300 dpi, if possible, the Authors should increase the quality of the blots

A. FINAL FILES:

B. MANUSCRIPT ORGANIZATION AND FORMATTING:

Sincerely,

Reviewer #1 (Comments to the Authors (Required)):

The authors have addressed all my concerns, therefore, it is acceptable in its current form.

Reviewer #2 (Comments to the Authors (Required)):

The authors have comprehensively addressed all of my comments and I think they convincingly demonstrate FTO's role in the regulation of Wnt signaling pathway. Although no further in vivo analysis has not been performed, I agree with the author's suggestions that discuss the physiological significance in the discussion section. I recommend the MS for publication.

Reviewer #3 (Comments to the Authors (Required)):

I think the authors resolved concerns raised by me appropriately.

Response to Reviewer

Reviewer #1 (Comments to the Authors (Required)):

The authors have addressed all my concerns, therefore, it is acceptable in its current form.

Reviewer #2 (Comments to the Authors (Required)):

The authors have comprehensively addressed all of my comments and I think they convincingly demonstrate FTO's role in the regulation of Wnt signaling pathway. Although no further in vivo analysis has not been performed, I agree with the author's suggestions that discuss the physiological significance in the discussion section. I recommend the MS for publication.

Reviewer #3 (Comments to the Authors (Required)):

I think the authors resolved concerns raised by me appropriately.

We are glad to hear that the reviewers' concerns were addressed. We would like to express our sincere appreciation to the editor and all the reviewers for their constructive advice to help improve the manuscript.

February 1, 2022

RE: Life Science Alliance Manuscript #LSA-2021-01250RR

Dr. Hyunjoon Kim
Institute for Basic Science
Center for RNA Research, IBS
Bldg 203-521, Seoul National University
1 Gwanangno, Gwanak-gu
Seoul 08826
Korea, Republic of (South Korea)

Dear Dr. Kim,

Thank you for submitting your Research Article entitled "The m6A(m)-independent role of FTO in regulating WNT signaling pathways". It is a pleasure to let you know that your manuscript is now accepted for publication in Life Science Alliance. Congratulations on this interesting work.

DISTRIBUTION OF MATERIALS:

Again, congratulations on a very nice paper. I hope you found the review process to be constructive and are pleased with how the manuscript was handled editorially. We look forward to future exciting submissions from your lab.

Sincerely,
